# Mulberry Leaf Extract Improves Metabolic Syndrome by Alleviating Lipid Accumulation In Vitro and In Vivo

**DOI:** 10.3390/molecules27165111

**Published:** 2022-08-11

**Authors:** Liangyu He, Yan Xing, Xinxiu Ren, Mengjiao Zheng, Shiqiang Yu, Yinbo Wang, Zhilong Xiu, Yuesheng Dong

**Affiliations:** 1School of Bioengineering, Dalian University of Technology, Dalian 116024, China; 2Dianxi Research Institute, Dalian University of Technology, Baoshan 678000, China

**Keywords:** metabolic syndrome, mulberry leaf extract, flavonoids, lipid metabolism, gut microbiota

## Abstract

Metabolic syndrome (MS) is a metabolic disease with multiple complications. Mulberry leaf extract (MLE) is rich in flavonoids and has great potential in alleviating glucose and lipid metabolism disorders. This study evaluated the effect and mechanism of MLE on the alleviation of MS. The components of the MLE were analyzed, and then the regulation of lipid metabolism by MLE in vitro and in vivo was determined. In a hepatocyte model of oleic acid-induced lipid accumulation, it was found that MLE alleviated lipid accumulation and decreased the expression of genes involved in lipogenesis. Furthermore, MLE improved obesity, insulin resistance, plasma lipid profile, and liver function in MS mice after a 15-week intervention. MLE decreased the expression of SREBP1, ACC, and FAS through the AMPK signaling pathway to inhibit lipid synthesis and increase the level of CPT1A to promote lipid decomposition to achieve its hypolipidemic effect. Meanwhile, MLE was also shown to affect the composition of the gut microbiota and the production of short-chain fatty acids, which contributed to the alleviation of lipid accumulation. Our results suggest that MLE can improve MS by improving lipid metabolism through multiple mechanisms and can be developed into dietary supplements for the improvement of MS.

## 1. Introduction

Metabolic syndrome (MS) is a syndrome that coexists with a series of risk factors including hyperglycemia, hyperlipidemia, and central obesity. MS is mainly characterized by a combination of multiple risk factors associated with metabolism. A large number of studies have shown that MS is closely associated with increased risk of cardiovascular diseases, cerebrovascular diseases, and type 2 diabetes mellitus (T2DM) [1]. In addition, MS and its related complications involve a wide range of disorders. MS breaks the balance of energy intake and energy consumption in the human body, and the rapid increase worldwide in the incidence of MS has seriously affected human health and changed people’s lifestyles [2]. Therefore, how to develop effective treatments of MS and related complications are problems that need to be urgently solved.

Insulin resistance (IR) is a physiological abnormality that targets tissues and exhibits decreased response sensitivity to insulin, which can cause abnormal blood glucose levels and lead to related physiological diseases. As the main feature of MS, IR is closely associated with glucose intolerance, dyslipidemia, hypertension, and endothelial dysfunction. This phenomenon is generally referred to as MS or insulin resistance syndrome, and its clinical features refer to obesity, abnormal plasma triglycerides and high-density lipoprotein cholesterol levels, and impaired glucose tolerance.

Metabolic syndrome is prone to lipid metabolism disorders. Fatty acid synthesis and oxidation are important processes in lipid metabolism, which involve the regulation of multiple transcription factors. Adenosine 5′-monophosphate (AMP)-activated protein kinase (AMPK) is an energy sensor that directly or indirectly regulates the mRNA expression or the activity of lipogenesis and lipolysis target proteins [3]. Especially, AMPK is involved in the regulation of sterol-regulatory element binding proteins-1 (SREBP1), a transcription factor involved in fatty acid synthesis [4]. SREBP1 regulates fatty acid synthesis by regulating the expression of genes encoding enzymes such as acetyl-CoA carboxylase (ACC) and fatty acid synthase (FAS) which mediate the de novo lipogenesis pathway [5]. Meanwhile, studies have shown that carnitine palmitoyl transferase 1A (CPT1A) is involved in the initiation of fatty acid β-oxidation. These transcription factors and enzymes together control the lipid metabolism of the body.

Mulberry leaves, i.e., the dried leaves of the edible plant *Morus alba* L., are widely used in medical and health care [6], and flavonoids are a group of the main active components of mulberry leaves. Previous studies have shown that flavonoids of mulberry leaves have anti-oxidation [7] and hepatoprotective activities [8]. Moreover, flavonoids of *Morus alba* L. have been shown to exhibit promising antiviral, antibacterial, and anti-inflammatory activities both in vivo and in vitro [9,10,11]. Owing to their outstanding biological properties, flavonoids have been shown to play major roles in lowering blood glucose level, blood lipid level, and free fatty acid levels, which were associated with MS [12]. Our group previously investigated the combinatorial effects of baicalein, a dietary flavonoid, and acarbose on prediabetes-associated nonalcoholic fatty liver disease, and found that inhibiting the de novo lipogenesis and decreasing lipid accumulation by baicalein its metabolites were the major mechanisms [13]. The improvements of mulberry leaves extract (MLE) on MS, including hyperglycemia and obesity, have been reported [14,15,16]. However, the mechanism of alleviating MS by MLE, especially, and mechanisms related to lipid accumulation, have not been well understood yet.

Gut microbiota is the general term for a variety of microorganisms that exist in the gut. The composition of gut microbiota is mainly divided into firmicutes, bacteroidetes, actinobacteria, and proteobacteria [17]. Studies have shown that the dysregulated gut microbiota is closely related to the occurrence of diabetes, obesity, atherosclerosis, and autism [18]. Therefore, the gut microbiota has an outstanding prospect as a research field related to metabolic syndrome. Current treatments targeting gut microbiota have been shown to improve metabolic function in humans [19,20], and recent studies have established the linkage between the lipid metabolism and metabolites of gut microbiota [21,22,23]. However, the effect of MLE on MS by regulating the gut microbiota mechanism has not been well-documented.

In this study, the active components of MLE were analyzed, and its effects on alpha mouse liver 12 (AML-12) cells and high-fat diet (HFD)/streptozotocin (STZ)-induced metabolic syndrome in C57BL/6 mice were evaluated. In addition, the mechanisms of MLE on lipid accumulation through de novo lipogenesis, fatty acid oxidation, and gut microbiota were also investigated. The aim of this work is to thoroughly investigate the lipid metabolism pathway regulated by MLE, which will lay a solid foundation for the development of functional foods for the improvement of MS.

## 2. Results

### 2.1. The Main Components in MLE

To understand the main components in MLE, the high-resolution LC-MS/MS data of the extracts were analyzed by the approach integrating computational programs MS-DIAL, MS-FINDER, and our group’s experienced web-based MetaboAnalyst tool [24]. A total of six identified compounds were summarized in Table 1. The data indicated that four of the compounds identified were flavonoids, including rutin (**3**), hyperoside (**4**), astragalin (**5**), and luteolin-7-glucoside (**6**). To further verify the identification, the three components with higher peak areas in HPLC (**3**–**5**) were compared with commercial standards, and the identities of the compounds were confirmed. The HPLC chromatogram of MLE generated by full-scan diode array detector (PDA) and the structures of identified components were shown in Figure 1A,B, respectively. These compounds have been identified in mulberry leaves and the flavonoids were major components in MLE [25,26].

### 2.2. The Effect of MLE on Lipid Accumulation in AML-12 Cells Induced by Oleic Acid

The cytotoxicity assay was performed to determine the appropriate dose of MLE and the appropriate concentration of oleic acid for subsequent studies. As shown in Figure 2A, the viability of the cells administered with MLE with a concentration above 50 μM was significantly increased compared to the control group. In order to reduce the interference of cell viability factor on the results, the experiments were designed to examine the effects of flavonoids at concentrations of 3.125, 6.25, 12.5, and 25 μM. Meanwhile, it was found that oleic acid at concentrations of 400 and 800 μM had significant cytotoxic effects on AML12 cells. In order to ensure the stability of the subsequent staining, oleic acid at a concentration of 200 μM was selected as the induction concentration of the lipid accumulation model (Figure 2B). The results showed that MLE significantly attenuated oleic acid-induced lipid accumulation in AML-12 cells (Figure 2D–I) in a dose-dependent manner (Figure 2C). The RNA level of leptin in the cells was significantly increased in a dose-dependent manner (Figure 2K), and the RNA level of adiponectin was also increased significantly with the increase of MLE (Figure 2J).

### 2.3. Effects of MLE on the Expression of Proteins Involved in Lipid Accumulation Induced by Oleic Acid in AML-12 Cells

The levels of target proteins were determined by immunoblotting. According to the results, the expression levels of ACC and FAS in the lipid accumulation model group were significantly increased compared with those in the control group, and their expression levels decreased significantly after the administration of MLE, showing a dose-dependent effect (Figure 3B,D). The CPT1A content in the model group was significantly decreased (*p* < 0.05), and MLE administration achieved a significant increase at the higher dose (12.5 μM) (Figure 3C).

### 2.4. Effects of MLE on Body Weight and Organ Weight of MS Mice

Compared to the control group, the MS model group significantly increased the body mass at Week 15 (Figure 4A), showing the role of MS in inducing obesity, and the administration of MLE may provide the effect of inhibiting weight gain. A high-fat diet increased the relative weights of organs such as epididymal fat, the liver, and the kidneys (Figure 4B–D), and after the intervention of MLE, the weight gain of organs such as epididymal fat, the liver and the kidneys was suppressed (*p* < 0.05).

### 2.5. The Effect of MLE on Plasma Glucose and Insulin Resistance in MS Mice on Week 15

The effect of MLE on plasma glucose and insulin resistance in MS mice was verified. Compared to the control group, the fasting plasma glucose and 2 h postprandial glucose (2 h-PG) levels were significantly increased in the MS mice, indicating impaired glucose tolerance. After the 15-week intervention, the levels of fasting plasma glucose and 2 h-PG of the mice receiving the treatment of MLE were significantly decreased (*p* < 0.05) (Figure 5A,B). OGTT and AUC results showed that MLE could improve the impaired glucose tolerance induced by MS (Figure 5C,D). Fasting insulin levels in HFD/STZ-induced MS mice increased significantly (Figure 5E). Meanwhile, the change of HOMA-IR index showed the same trend as the results above (Figure 5F), indicating that the insulin sensitivity was impaired in MS. The administration of MLE decreased the fasting insulin level and HOMA-IR index and these results reached the same level as the positive control (metformin) group and the normal group, which indicates that the MLE improved the physiological abnormalities caused by MS.

### 2.6. The Effect of MLE on Plasma Lipid Profile and Hepatic Inflammatory Factors in HFD/STZ-Induced Mice

Abnormal lipid metabolism in MS is generally manifested as elevated levels of total cholesterol, triglyceride, and low-density lipoprotein cholesterol (LDL-C) in the plasma, with decreased levels of high-density lipoprotein cholesterol (HDL-C). As shown in Figure 6, compared to the MS model group, the MLE decreased the levels of total cholesterol and triglyceride in the serum (Figure 6A,B; *p* < 0.05). Meanwhile, MLE significantly increased the level of HDL-C, and decreased the level of LDL-C compared to the model group (Figure 6C,D; *p* < 0.05). Furthermore, it was found that the levels of ALT and AST in MS mice were significantly increased (Figure 6E,F) (*p* < 0.05), indicating impaired liver function. The levels of ALT and AST in the experimental group were significantly decreased after the MLE intervention, indicating that the treatment effect was significant.

### 2.7. Histological Examination of Liver Tissue and Epididymal Adipose Tissue

In order to test the hypothesis whether MLE can alleviate liver injury in the MS model, we performed HE staining to detect the pathological changes in the liver tissue and epididymal adipose tissue of mice after 15 weeks of administration. The results showed that the MLE decreased the size and number of lipid vacuoles and alleviated the phenomenon of loose liver tissue in the liver, inhibited the lipid accumulation in the liver, and attenuated hepatocyte degeneration induced in the MS model (Figure 7A–E and Appendix A).

The effect of MLE on lipid accumulation was also determined in HE-stained epididymal adipose tissues. Compared to the control group, elevated lipid accumulation was observed in the adipose tissues of the model group, as evidenced by the increased size of the adipocytes. Following the treatment of MLE, the size of adipocytes was notably decreased, reaching levels close to the metformin group (Figure 7F–J and Appendix A).

### 2.8. The Effect of MLE on Lipid Synthesis and Decomposition in MS Mice

The regulatory effect of MLE on the lipid metabolism pathway of MS was further determined. As shown in Figure 8, after the model group was treated with MLE, the level of phosphorylated AMPK in the liver was significantly increased (Figure 8A,B), and the levels of SREBP1 and ACC in the liver were significantly decreased, returning to the same level as the metformin group, suggesting that MLE has an inhibitory effect on lipid synthesis (Figure 8A,C,D). The intervention of MLE also significantly increased the level of CPT1A (Figure 8A,E), and the increase of CPT1A contributes to the decomposition and oxidation of fatty acids. Compared to the model group, MLE significantly decreased the level of FAS in the liver and decreased lipid synthesis (Figure 8A,F), which was consistent with the effect of reducing the level of ACC. MLE regulated the abnormal lipid metabolism symptoms of MS by regulating the expression of AMPK and the related transcription factors, promoting fatty acid decomposition and oxidation and inhibiting lipid synthesis.

### 2.9. The Effects of MLE on the Structure of Gut Microbiota and the Types and Contents of SCFAs in MS Mice

MS significantly changed the structure and composition of the gut microbiota, and the treatment of MLE alleviated the imbalance of the abundance of some gut microbiota, and also improved the gut microbiota by promoting the changes of others (Figure 9A). At the phylum level, compared to the control group, the abundance of firmicutes in the MS group was significantly increased, and the abundance of bacteroides was significantly decreased. After the administration of MLE, the abundance of firmicutes was significantly decreased, the abundance of bacteroides was increased to a certain extent, and the ratio between the two phyla showed a great difference (Figure 9B). Meanwhile, after the administration of MLE, the abundance of *Lactobacillus*, *Bifidobacterium,* and *Allobaculum* was significantly increased, compared to the model group. (*p* < 0.05) (Figure 9C–E). The abundance of *Helicobacter* was significantly decreased, compared to the model group. (*p* < 0.05) (Figure 9F) The genus-level clustering heat map showed that the MS group had a greater impact on the content of gut microbiota, and after MLE intervention, the content of some bacterial genera showed a certain trend of recovery (Figure 9G).

SCFAs as microbial metabolites are mainly derived from the intestinal fermentation of dietary fiber and play an important role in regulating circadian rhythms and metabolic functions. Therefore, the content of SCFAs in the feces of different groups was determined by gas spectrometry. As shown in Figure 10, the level of SCFAs were significantly decreased in the model group, which was correlated to the dysregulated gut microbiota. Compared to the model group, the levels of SCFAs were significantly increased in the MLE group (Figure 10). For propionic acid and isobutyric acid, the levels in the MLE group were increased to a comparable level to the control group (Figure 10B,D). For acetic acid and total SCFAs, the levels in the MLE group were even higher than those in the control group (Figure 10A,E). These data showed that MLE significantly increased the content acetic acid, propionic acid, butyric acid, isobutyric acid, and total SCFAs compared to the model group, thereby improving the intestinal microenvironment (Figure 10).

## 3. Discussion

MS involves a variety of risk factors related to metabolism, including hyperglycemia, dyslipidemia, and obesity. Moreover, MS also leads to the development of multiple associated complications. The current pharmacological treatment of MS focuses on oral antidiabetic drugs, insulin drugs, hypertension drugs, statins/fibrates, and anti-obesity drugs. Long-term drug treatment can easily lead to side effects and drug resistance, and the use of natural products can play a role in adjuvant therapy [27]. Mulberry leaves have been demonstrated as a natural product with hypoglycemic and hypolipidemic effects. In this study, we found that MLE improved MS, and explored its mechanism of action in abnormal lipid metabolism. The results showed that MLE improved obesity, insulin signaling, abnormal lipid metabolism, liver damage in vivo, and improved lipid accumulation in vitro. Further research found that MLE improved lipid accumulation by inhibiting lipid synthesis and promoting lipolysis. At the same time, we found that MLE also improved glucose-lipid metabolism disorders by affecting the composition of gut microbiota. Our study provided a new theoretical basis for the development of MLE for the adjuvant treatment of MS.

IR is an important feature in MS, which is also an important factor leading to metabolic disorders. Our study found that MLE effectively improved HOMA-IR in HFD/STZ-induced MS mice, and the levels of AUC and OGTT have also been significantly improved. Existing studies have demonstrated the feasibility of using HOMA-IR and OGTT as indicators of MS [28]. These results suggest that MLE can improve MS in HFD/STZ-induced mice.

As an important factor of MS, IR is also closely related to central obesity, and the presence of IR and obesity is more likely to cause abnormal lipid metabolism in the body. We did find associated obesity phenomenon in our characterization of MS mice, and significant improvement was also seen after MLE intervention (Figure 1 and Figure 5). Furthermore, our results confirmed that the levels of TG, T-CHO, HDL-C, and LDL-C all changed significantly in MS mice. Similar work was done by Zhang et al., whose results were more focused on comparing the effects of different flavonoids on metabolic syndrome [29]. In contrast, our research is more focused on in-depth exploration of the underlying mechanism. After the intervention of MLE, the indicators of lipid metabolism were improved and recovered to a certain extent. These results suggest that MLE can improve obesity symptoms and abnormal lipid metabolism in MS mice.

In addition, lipid synthesis and oxidation are mainly carried out in hepatocytes, and it has been reported that a large number of experimental studies and comparisons have found that metabolic disorders are frequently accompanied by liver damage, which shows that liver damage plays an important role in metabolic disorders [30]. Therefore, we explored whether the abnormal lipid metabolism in MS mice was caused by hepatocytes damaging. The results showed that ALT and AST indicators, which was used to measure the degree of liver function, were significantly increased in the model group and recovered after MLE intervention, indicating that the liver is indeed damaged to a certain extent during metabolic disorders. The results of liver histology further confirmed this observation (Figure 7). Meanwhile, numerous lipid vacuoles were identified in the MS group, indicating that the characteristics of non-alcoholic fatty liver appeared. There were existing reports indicating that non-alcoholic fatty liver disease is a key factor leading to MS [31]. At this stage, we cannot confirm whether MS is caused by non-alcoholic fatty liver disease. It is possible that non-alcoholic fatty liver disease and MS are closely related to each other and are likely to be triggered at the same time. Some studies have explored the effect of related drugs at the cellular level [32,33]. Based on the research above, we investigated the effect of MLE on oleic acid-induced lipid accumulation in AML12 cells. Our results showed that MLE effectively alleviated the degree of lipid accumulation in mouse hepatocytes, which was dose-dependent. Due to the same trend shown in vivo, we speculate that some active substances may be absorbed into the body during the metabolic process to exert their effects. Li et al. also showed that the treatment of natural products has a great potential on the alleviation of MS [34]. Leptin and adiponectin play regulatory roles in glycolipid metabolism, and their deficiency leads to increased free fatty acids and accumulation of hepatic lipids. Then, we explored the effect of MLE at a deeper level on lipid accumulation in AML-12 cells, and the results showed that RNA levels of leptin and adiponectin were increased, which was consistent with the trend reported in the literature [35,36]. Changes in the expression levels of lipid metabolism-related proteins in cells showed that the effect of MLE went deep into the protein level. In conclusion, MLE can improve MS by regulating lipid metabolism disorder, thereby protecting liver tissue.

Lipid metabolism is an important metabolic process in the liver, mainly including fatty acid synthesis and oxidation [37]. Existing results showed that MLE significantly improved blood lipid indexes. In vitro oil red O staining and histological staining in vivo also showed that MLE treatment improved lipid accumulation. Therefore, we explored the regulators involved in the regulation of lipid metabolism. Lipid metabolism is influenced by AMPK, which in turn regulates SREBP1, a key protein involved in fatty acid synthesis and oxidation. At the same time, studies have shown that the downstream protein FAS is involved in catalyzing fatty acid synthesis, ACC is involved in inhibiting fatty acid oxidation [38], and CPT1A is involved in catalyzing fatty acid oxidation [39]. Our results showed that AMPK and SREBP1 were significantly down-regulated after MLE intervention, whereas the downstream proteins FAS and ACC also showed a significant down-regulation trend, and CPT1A was significantly recovered after treatment. Based on the existing results and related literatures, we propose that MLE regulates lipid metabolism by the pathway of regulating the AMPK center, downregulating the key protein of SREBP1, and then promoting fatty acid oxidation and inhibiting fatty acid synthesis. Liang et al. explored the effect on lipid accumulation in AML-12 cells, and their results also showed that AMPK can regulate downstream pathways as a central regulator [40]. The difference is that their focus is more on the role of AMPK in regulating autophagy and apoptosis. We put more emphasis on how AMPK phosphorylation regulates downstream lipid oxidation and synthesis pathways to achieve its effect. The focus of Foret et al.’s study is that AMPK activation can reduce the content of lipid in liver by increasing fat oxidation in the body, which is consistent with the trend of our results [41]. Meanwhile, we also explored its effect on increasing lipid β-oxidation and inhibiting fat synthesis. Hu et al. explored the effect of total flavonoids from mulberry leaves on liver cholesterol metabolism disorders in NAFLD rats. The results showed that MLE significantly down-regulated the expression levels of lipid-related markers such as SREBP2, HMGCR, miR-33a, and CYP7A1 in the liver, inhibited lipid accumulation in a rat model of NAFLD, and attenuated liver injury, thereby exerting a protective effect during NAFLD treatment [42]. Our results not only showed the effect of MLE on lipid metabolism-related markers including SREBP and CPT1A, but also directly demonstrated the direct effect of MLE by promoting lipid oxidation and decomposition through the expression of downstream-related genes such as FAS, which was verified by both in vitro and in vivo. Li et al. studied the efficacy and mechanism of alkaloids, flavonoids, polysaccharides, and other bioactive components in mulberry leaves alone in the treatment of T2DM [43]. UPLC-QTOF/MS was used to evaluate the differential metabolism from various metabolic pathways by urine metabolomics. The related pathway changes were explored through HepG2 cells and 3T3-L1 cells. The results showed that 1-DNJ and isoquercitrin (QG) increased glucose uptake and promoted cell differentiation in HepG2 cells through AGEs/RAGE and p38 MAPK/NF-κB pathways, and increased PPARγ, C/EBPα, and SREBP-1 in 3T3-L1 cells expression of GLUTag inhibits age-induced GLUTag cell damage and apoptosis. We also conducted research at the cellular level and used in vivo studies to explore and verify related pathways and mechanisms, and we were more inclined to understand the effect of MLE on lipid metabolism. Taken together, these findings suggest that MLE intervention can improve lipid metabolism disorders by regulating lipid metabolism (inhibiting lipid synthesis and promoting lipolysis) through the AMPK signaling pathway.

The causes of MS are complex and diverse, and dietary habits are also one of the influencing factors [44]. A growing body of research suggests that diet can influence the gut microbiome, which play a role in promoting energy metabolism. Our study found that the microbiota composition of MS mice showed a significant decrease in bacteroidetes, a significant increase in firmicutes, and a significant increase in the abundance of proteobacteria with changes in diet. The abundance of *proteobacteria* is an important marker of microbial community instability [17]. After the intervention of MLE, the abundance of relevant microbiota recovered to a certain extent, which was consistent with the literature [45]. We also analyzed the diversity of gut microbiota species, and the results showed that the number of gut microbiota species in the model group and MLE-intervention group was significantly higher than that in the normal group, and the results were different from the literature [46]. The cluster heatmap showed that the intervention of MLE led to changes in the content of relevant gut microbiota. For instance, the relative abundance of *Allobaculum* decreased in the model group. In contrast, the relative abundance of *Allobaculum* was higher than that in the normal group after the intervention of MLE. It has been reported that *Allobaculum* is a producer of SCFAs, which is negatively correlated with obesity [47,48,49]. The same trend was reflected in *Lactobacillus* and *Bifidobacterium*. In contrast, *Helicobacter* is prone to the production of endotoxin, which is a hallmark of gastric patients [50,51]. Our results are consistent with those of the related literature [42], which can effectively illustrate that the intervention of MLE can achieve a new balance of gut microbiota by restoring beneficial microbiota and inhibiting harmful microbiota and has a certain regulatory effect on lipid metabolism. The metabolites expressed by the gut microbiota also changed accordingly. Among them, SCFAs are important metabolites produced by the gut microbiota, which are produced by breaking down dietary fibers, proteins, and peptides [52]. SCFAs affect the corresponding metabolic functions of the body by entering the blood, among which the most abundant SCFAs are acetate, propionate, and butyrate. Previous studies have found that SCFAs play a certain role in glucose and lipid metabolism disorders. Acetate inhibits lipid accumulation by promoting lipid decomposition and fatty acid oxidation and inhibiting fatty acid synthesis [22]. Propionic acid indirectly causes AMPK activation, which leads to the inhibition of gluconeogenesis [23]. Butyrate improves energy metabolism indirectly by reducing energy intake and activating brown adipose tissue to promote lipid oxidation [21], suggesting that butyrate may be involved in weight change. Our experimental results showed that the intervention of MLE increased the content of SCFAs (Figure 8). For instance, the content of acetate propionate in the MLE-intervention group showed an upward trend, and the 16S rRNA gene sequencing results also showed that the proportion of bacteroidetes in the gut microbiota increased. As bacteroidetes mainly produce acetate and propionate, we speculate that MLE may regulate lipid metabolism and improve MS by regulating gut microbiota and then regulating the content of SCFAs produced by the gut microbiota, which is also consistent with the results pf previous lipid metabolism characterization studies. The level of SCFAs increased after the treatment of MLE, indicating that MLE-intervention has a positive effect on the abundance and metabolites (SCFAs) of gut microbiota, which is helpful to alleviate the glucose and lipid metabolism disorders caused by MS. The development of dietary supplements based on MLE for the improvement of MS is an interesting topic to be explored in the future.

## 4. Materials and Methods

### 4.1. Materials and Reagents

Mulberry leaves were collected from Longling county (Baoshan, China) and identified to be the leaves of *Mours alba* by Prof. Baolian Guo, the Institute of Medicinal Plant Development, Peking Union Medical College. The insulin detection kit was purchased from Wuhan Miting Biotechnology Co., Ltd. (Wuhan, China). Assay kits for triglyceride (TG), total cholesterol (T-CHO), high-density lipoprotein cholesterol (HDL-C), low-density lipoprotein cholesterol (LDL-C), alanine aminotransferase (ALT), and aspartate aminotransferase (AST) were purchased from Jiancheng Bioengineering Institute (Nanjing, China). Streptozotocin was purchased from Sigma Co., (Saint Louis, MO, USA). Standard mouse diet was purchased from Mouse One Mouse Two Biological Co., Ltd. The antibodies for AMPK, SREBP1, ACC, CPT1A, FAS, and horseradish peroxidase-conjugated goat-anti-rabbit IgG were purchased from Beyotime Biotechnology Co., Ltd. (Shanghai, China). Other chemicals were supplied by Damao Co., Ltd. (Tianjin, China). Standards of rutin, hyperoside, and astragalin were purchased from Chengdu Must Bio-Technology Co., Ltd. (Chengdu, China).

### 4.2. Preparation of MLE

MLE was prepared according to the procedure reported in the previous literature [53], and the detailed procedure was as follows: The mulberry leaves were dried and soaked in 80% ethanol solution for extraction at room temperature for 12 h. The 80% ethanol extraction was filtered to remove the dregs, and the solvent was evaporated. Subsequently, petroleum ether was added to the obtained samples. The organic phase was removed by a liquid phase separation method, then ethyl acetate was added to the aqueous phase for extraction. The ethyl acetate phase was collected, dehydrated with anhydrous sodium sulfate, and evaporated to dryness to obtain MLE.

### 4.3. Cell Culture and Treatments

Alpha mouse liver 12 (AML-12) cells were purchased from Shanghai Stem Cell Bank. AML-12 cells were cultured in Dulbecco’s modified Eagle Medium (DMEM) supplemented with 10% fetal bovine serum, 100 IU/mL penicillin, and 100 μg/mL streptomycin (Keygen, Nanjing, China) and incubated at 37 °C in 5% CO_2_. The establishment of the lipid accumulation model was performed according to the previous literature [33] as follows: Briefly, AML-12 cells were seeded overnight at 2 × 10^5^ cells/well in a 12-well plate. The next day, different concentrations (0, 100, 200, 400 and 800 μM) of oleic acid (OA, Sigma-Aldrich, Saint Louis, MO, USA) were added to the wells. The cells were then incubated for another 24 h, and the lipid accumulation was detected by Oil Red O (ORO) staining. After the model was established, the follow-up administration operations were as follows: the cells were treated with MLE at different concentrations (0, 3.125, 6.25, 12.5 and 25 μM, respectively). After 24 h, the viability of the cells was analyzed using the Cell Counting Kit-8 (CCK-8) detection kit (Shizebio, Shanghai, China). Alternatively, the lipid accumulation in cells was detected by ORO staining, and the images were acquired with an IX83 inverted microscope (Olympus Co., Tokyo, Japan). The percentage of stained area in each image was determined using ImageJ software (version 1.8.0, National Institute of Healt, Bethesda, MD, USA).

### 4.4. Animals and Experimental Protocols

The metabolic syndrome mice model was established according to the previous literature [42]. C57BL/6 mice (body mass 18–22 g, *n* = 32) were purchased from Changsheng Biotechnology Co., Ltd. (BenXi, Liaoning, China). All animal experiments were approved by the Institutional Animal Care and Use Committee of Dalian University of Technology and were carried out in strict accordance with the standards of institutional animal ethics. The institutional protocol code is 2020-021. In this study, the mice were housed in an environment with a temperature of 22.2 °C and a relative humidity of 55.5%, and received a 12 h:12 h light–dark cycle (lights on at 08:00). Animal acclimation was performed 1 week before the experiment. The metabolic syndrome mice model was induced by HFD and STZ according to the previous protocol [28,54]. Oral glucose tolerance test (OGTT) was performed on Days 7, 14, and 21 after streptozotocin injection, respectively. Mice conforming to the criteria of MS model were randomized into 3 groups (*n* = 8), and were administered with 0.9% NaCl (model group), metformin (200 mg/kg/d, Met group), or MLE (100 mg/kg/d, MLE group) for 15 weeks. Age-matched normal littermates (*n* = 8) were used as controls.

### 4.5. Sample Collection and Storage

Sample processing of each group was performed according to a protocol by Zhou et al. [54]. After the drug intervention, the mice were euthanized by anesthesia, and the plasma and liver tissue were collected and stored at −80 °C until use. The feces and cecal contents were collected under aseptic conditions to analyze the content of short-chain fatty acids (SCFAs) and the composition of the gut microbiota. The relative organ weight was calculated as the ratio of the organ weight to the body weight. Part of the fresh liver specimens and epididymal adipose tissue were fixed in 4% paraformaldehyde for histology analysis.

### 4.6. Oral Glucose Tolerance Test

The oral glucose tolerance test was performed according to the previous literature [13]. The mice were fasted with free access to water for 14 h, and each mouse was administered with 2 g/kg glucose by oral gavage. Plasma glucose levels of mice were measured by tail vein blood sampling at 0 min (PG_0 min_), 30 min (PG_30 min_), 60 min (PG_60 min_), 90 min (PG_90 min_), and 120 min (PG_120 min_) after glucose oral gavage. The area under the plasma glucose curve (AUC) was calculated using the following equation:AUC = (PG_0 min_ + 2 × PG_30 min_ + 2 × PG_60 min_ + 2 × PG_90 min_ + PG_120 min_) × 30 min × 0.5(1)

### 4.7. Determination of Serum Biomarkers and HOMA-IR

The levels of the lipid profile and fasting insulin were determined with the corresponding assay kit according to the manufacturer’s instructions (Jiancheng Biotechnology Institute, Nanjing, China). The homeostasis model assessment (HOMA)-insulin resistance (HOMA-IR) index was calculated according to the formular reported in the existing literature as follows [55]:HOMA-IR = fasting insulin level (µIU/mL) × fasting glucose level (mM)/22.5(2)

### 4.8. Histopathological Observation

The liver and epididymal fat of mice in each group (*n* = 4) were collected and fixed with 4% paraformaldehyde for 24 h. The fixed tissues were dehydrated with ethanol gradient, vitrified in xylene, and embedded in paraffin. Sections were 5 μM and stained with hematoxylin-eosin (HE). Stained sections were observed with an IX83 inverted microscope (Olympus Co., Tokyo, Japan), and images were collected and recorded.

### 4.9. Western Blot Analysis

Liver tissue samples were homogenized on ice in radioimmunoprecipitation assay (RIPA) lysis buffer containing protease inhibitors cocktail and 1 mM phenylmethylsulfonyl fluoride (PMSF), and the proteins of interest were detected by Western blot as described previously [56]. The protein concentration of the lysate was detected by a bicinchoninic acid (BCA) protein assay kit. A total of 120 μg of protein was loaded per sample and subjected to sodium dodecyl sulfate polyacrylamide gel electrophoresis (SDS-PAGE). Equal amounts of proteins were separated on 12% sodium dodecyl sulfate-polyacrylamide gels and transferred to polyvinylidene fluoride (PVDF) membranes by electrophoretic transfer. After blocking in 5% nonfat milk in Tris-buffered saline with 0.05% Tween-20 (TBST) for 1 h, the membrane was incubated with primary antibodies overnight at 4 °C, and the primary antibodies used in this experiment included AMPK (1:1000), CPT1A (1:1000), SREBP1 (1:1000), FAS (1:1000), and ACC (1:1000). Subsequently, the PVDF membrane was washed 3 times in TBST and incubated with the appropriate secondary antibody (diluted by 1:1000) for 2 h. The membranes were then washed with TBST 3 times and then developed with enhanced chemiluminescence reagent and the bands were recorded by photography. The relative amount of protein was normalized using β-actin (Beyotime Biotechnology Co., Ltd., Shanghai, China) as the loading control.

### 4.10. Fecal Microbiota 16S rRNA High-Throughput Sequencing

The 16S rRNA sequencing and analysis was performed according to the previous literature [57]. Briefly, the microbial DNA was extracted using a genomic DNA isolation kit (OmegaBioTek, Doraville, GA, USA). The samples (*n* = 3) were handed over to Hangzhou Guhe Information Technology Co., Ltd. (Hangzhou, China) to amplify and sequence the V4 region using Illumina NovaSeq, followed by related data processing. After the total DNA of fecal samples was extracted, the samples were subjected to quality control and PCR amplification. After the amplification was completed, electrophoresis was performed, and the PCR products were qualified. The amplicons were ligated and the library was constructed and quality-controlled. The sequencing data were used for species identification analysis.

### 4.11. Fecal SCFA Measurement

The short-chain fatty acids in stool samples (*n* = 3) were quantified as previously described with minor modifications [56]. Briefly, total of 0.2 g of fresh feces was extracted with 1.6 mL of ethanol. The mixture was allowed to set at room temperature for 10 min and centrifuged at 12,000× *g* at 4 °C for 15 min. The supernatant was collected and filtered through a 0.22 µm filter for loading. The SCFA standards were prepared following the same procedure, followed by gas chromatography (GC) detection. The SCFAs in stool include acetate, propionate, butyrate, isobutyrate, and total SCFAs. An Agilent FFAB gas chromatography column (30 m × 0.32 mm, 0.5 μm) was used. The inlet temperature was 250 °C, the split ratio was 10:1, and the temperature program was programmed as follows: the initial temperature was 80 °C, maintained for 2 min and increased to 180 °C at 10 °C/min. The detector was a flame ionization detector (FID) and the injection volume was 5 μL.

### 4.12. Statistical Analysis

The results of this study are expressed as mean ± SEM. The experimental data were subjected to the Duncan test using SPSS 13.0 software (IBM Co., Armonk, NY, USA). One-way ANOVA was used to evaluate the statistical significance of data between groups, and *p* < 0.05 was considered as statistically significant.

## 5. Conclusions

In conclusion, we found that MLE improved MS in mice and explored its mechanism of action in abnormal lipid metabolism (Figure 11). The results showed that MLE improved obesity, decreased insulin resistance, improved lipid metabolism, decreased liver damage and fatty liver in vivo, improved liver function, and improved lipid accumulation in hepatocytes. The signaling pathway regulated lipid metabolism by inhibiting lipid synthesis and promoting lipolysis to improve lipid metabolism disorders. At the same time, our study found that MLE can also improve the effect of glucose-lipid metabolism disorder by affecting the composition of gut microbiota. To sum up, MLE, as a natural active component of mulberry leaves, is a great alternative for the adjuvant treatment of metabolic syndrome, and the means of prevention and treatment of metabolic syndrome based on a variety of natural products will be a direction full of interest and potential.

## Figures and Tables

**Figure 1 molecules-27-05111-f001:**
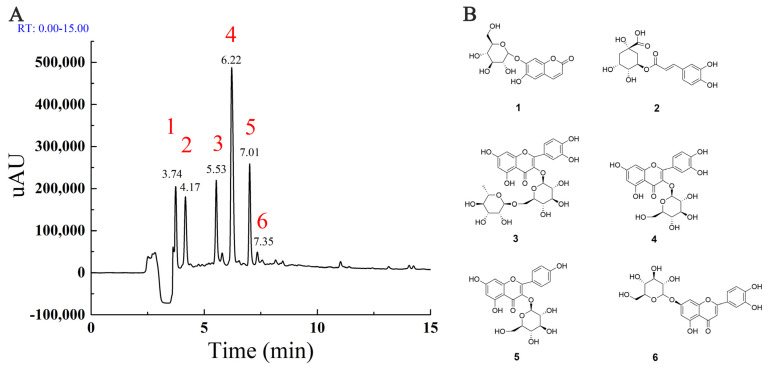
The major components in MLE (**A**): The HPLC chromatogram generated by full-scan PDA; (**B**): The structure of major components identified by HPLC-MS/MS.

**Figure 2 molecules-27-05111-f002:**
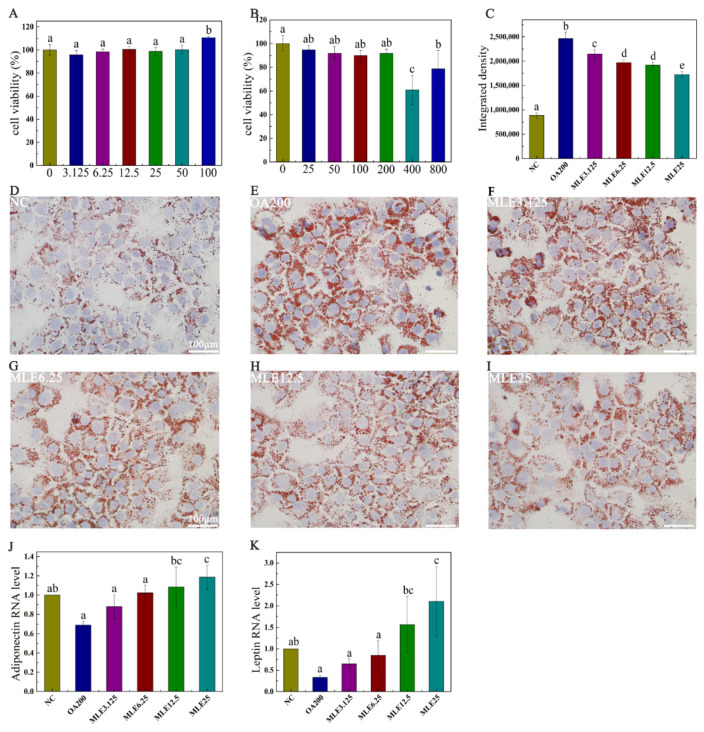
Effect of MLE on lipid accumulation induced by oleic acid in AML-12 cells. Cytotoxicity test of MLE at doses of 0 to 100 μM (**A**). AML-12 cells were cultured to 80% confluency and treated with dimethyl sulfoxide (DMSO, 0 μM) and lipid accumulation model (25–800 μM) at different oleic acid concentrations for cytotoxicity assay (**B**). When the confluency reached 80%, the cells were treated with DMSO (control), 200 μM of OA (OA200), and 200 μM of oleic acid with 3.125 μM, 6.25 μM, 12.5 μM and 25 μM of MLE, respectively. The cells were incubated at 37 °C for 24 h. Subsequently, these cells were washed with PBS and stained with oil red O (**D**–**I**). The RNA levels of adiponectin (**J**) and leptin (**K**) in cells after MLE intervention were also detected. The integrated density of ORO staining within each field was determined using ImageJ (version 1.8.0), and the data (*n* = 3) was expressed as the mean ± standard deviation (**C**). The results were analyzed by one-way ANOVA followed by Duncan test. Different letters (a–e) indicate values with statistically significant differences, “ab” indicates no significant difference from groups with letter “a” or “b”, “bc” indicates no significant difference from groups with letter “b” or “c”. Scale bar: 100 μM.

**Figure 3 molecules-27-05111-f003:**
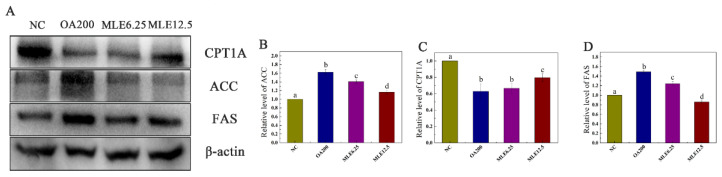
Effects of MLE on the expression of related proteins in lipid accumulation induced by oleic acid in AML-12 cells. The cells were divided into control group (Control), model group (OA200), and MLE administration group (MLE6.25 and MLE12.5). The cell lysates were analyzed by SDS-PAGE followed by immunoblotting with the indicated antibodies (**A**). Immunoblotting results were quantified by densitometry (**B**–**D**). The results were analyzed by one-way ANOVA followed by Duncan test. Different letters (a–d) indicate values with statistically significant differences.

**Figure 4 molecules-27-05111-f004:**
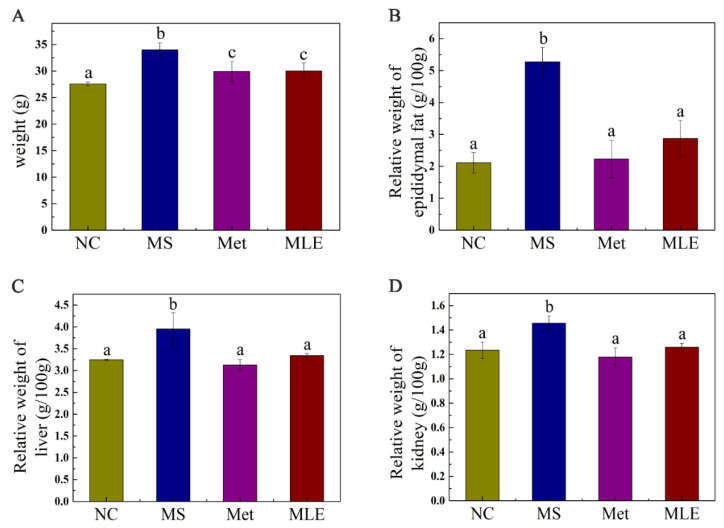
Effects of MLE on body weight and organ weight of MS mice. Control mice were administered with 0.9% NaCl (NC), HFD/STZ-induced MS mice were administered with 0.9% NaCl (MS), metformin (200 mg/kg/d, Met) and MLE extract (100 mg/kg/d, MLE) by oral gavage for 15 weeks, respectively. Effects of MLE on weight gain (**A**), epididymal fat (**B**), liver (**C**), kidney (**D**) relative organ weight after 15 weeks of intervention were determined. The results were analyzed by one-way ANOVA followed by Duncan test. Different letters (a–c) indicate values with statistically significant differences.

**Figure 5 molecules-27-05111-f005:**
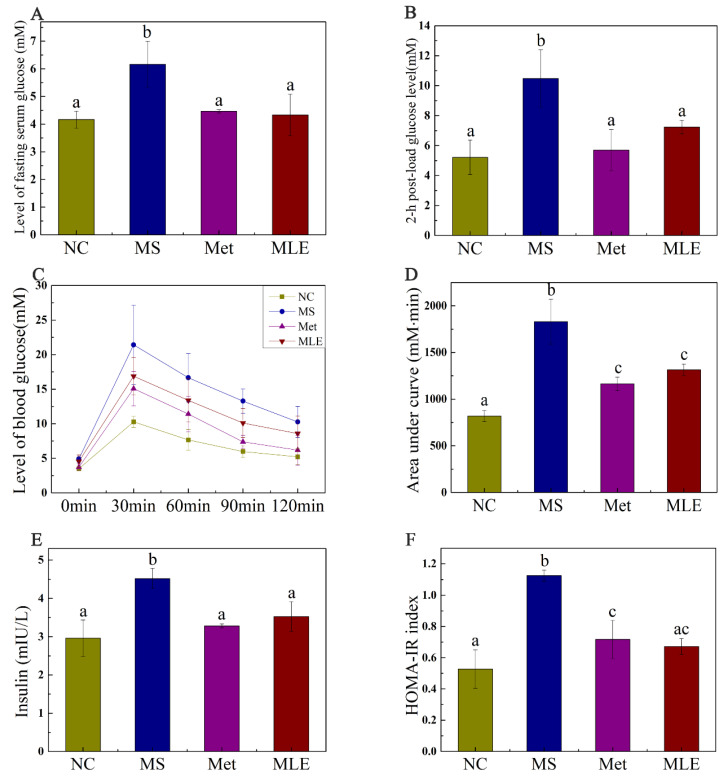
Effects of MLE on plasma glucose and insulin resistance in HFD/STZ-induced MS mice on Week 15. The mice in the control group were administered with 0.9% NaCl (NC), whereas HFD/STZ-induced mice were administered with 0.9% NaCl (MS), metformin (200 mg/kg/d, Met), and MLE (100 mg/kg/d, MLE) by oral gavage for 15 weeks. After the 15-week intervention, fasting plasma glucose level (**A**), 2-h postprandial plasma glucose level (**B**), 2-h plasma glucose profile (**C**), area under the plasma glucose profile (**D**), fasting insulin level (**E**), HOMA-IR index (**F**) were determined. Results were analyzed by one-way ANOVA followed by Duncan test. Different letters (a–c) indicate values with statistically significant differences, “ac” indicates no significant difference from groups with letter “a” or “c”.

**Figure 6 molecules-27-05111-f006:**
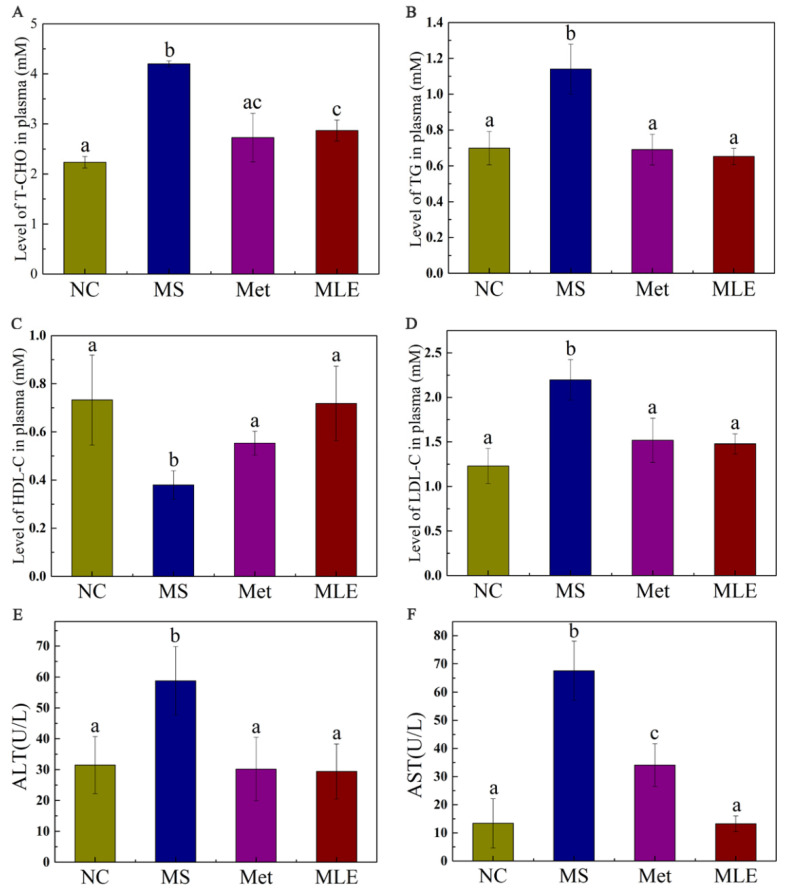
Effects of MLE on plasma lipid profiles and hepatic inflammatory factors in MS mice. The mice in the control group were administered with 0.9% NaCl (NC), whereas MS mice were administered with 0.9% NaCl (MS), metformin (200 mg/kg/d, Met) and MLE (100 mg/kg/d, MLE) by oral gavage for 15 weeks. After 15 weeks of intervention, the levels of serum total cholesterol (**A**), triglyceride (**B**), high-density lipoprotein cholesterol (**C**), low-density lipoprotein cholesterol (**D**), alanine aminotransferase (**E**), aspartate aminotransferase (**F**) in each group were determined. The results were analyzed by one-way ANOVA followed by Duncan test. Different letters (a–c) indicate values with statistically significant differences, “ac” indicates no significant difference from groups with letter “a” or “c”.

**Figure 7 molecules-27-05111-f007:**
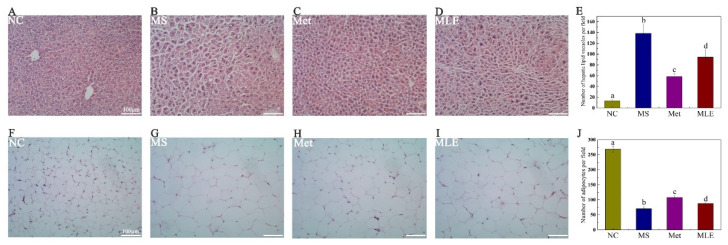
The effect of MLE on liver tissue and epididymal adipose tissue. The mice in the control group were administered with 0.9% NaCl (NC), whereas MS mice were administered with 0.9% NaCl (MS), metformin (200 mg/kg/d, Met), and MLE (100 mg/kg/d, MLE) by oral gavage for 15 weeks. After the intervention, the tissues of each group of mice were collected and fixed in 4% paraformaldehyde. (**A**–**D**) HE staining of liver tissue, (**E**) Number of hepatic lipid vacuoles per field in each group, (**F**–**I**) HE staining of epididymal adipose tissue, (**J**) Number of adipocytes per field in each group. Different letters (a–d) indicate values with statistically significant differences. Scale bar: 100 μM.

**Figure 8 molecules-27-05111-f008:**
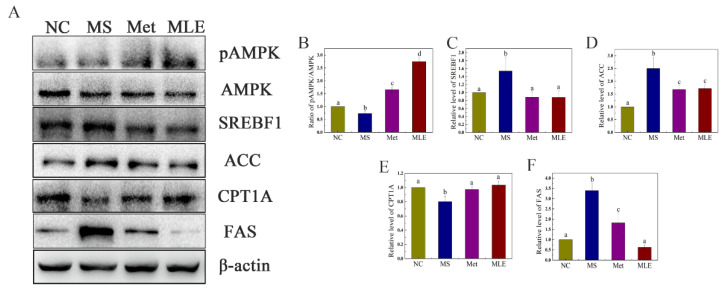
Effects of MLE on the decomposition of lipogenesis in MS mice. The mice in the control group were administered with 0.9% NaCl (NC), whereas MS mice were administered with 0.9% NaCl (MS), metformin (200 mg/kg/d, Met), or MLE (100 mg/kg/d, MLE) by oral gavage for 15 weeks. The liver tissue of each group of mice was homogenized. Liver lysates were analyzed by SDS-PAGE followed by immunoblotting with the indicated antibodies (**A**). The immunoblot results were quantified by densitometry analysis (**B**–**F**). The results were analyzed by one-way ANOVA followed by Duncan test. Different letters (a–d) indicate values with statistically significant differences.

**Figure 9 molecules-27-05111-f009:**
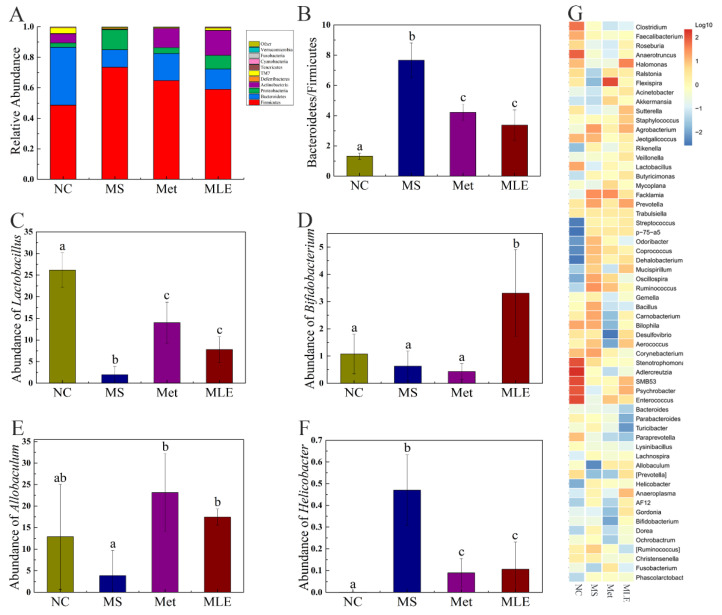
Effects of MLE on the structure of gut microbiota and the types, contents of SCFAs in HFD/STZ-induced mice. After 15 weeks of intervention, 16S rDNA gene sequencing and bioinformatics analysis were performed to determine the fecal microorganisms in the normal group (NC), model group (MS), metformin treatment group (Met), and MLE treatment group (MLE). (**A**) Taxonomic analysis of gut bacteria in different groups of mice at the phylum level. (**B**) The ratio of Firmicutes to Bacteroidetes in each group. Relative abundance of the *Lactobacillus* (**C**) and *Bifidobacterium* (**D**), *Allobaculum* (**E**), and *Helicobacter* (**F**) in each group. (**G**) Genus level clustering heatmap. The results were analyzed by one-way ANOVA followed by Duncan test. Different letters (a–c) indicate values with statistically significant differences, “ab” indicates no significant difference from groups with letter “a” or “b”.

**Figure 10 molecules-27-05111-f010:**
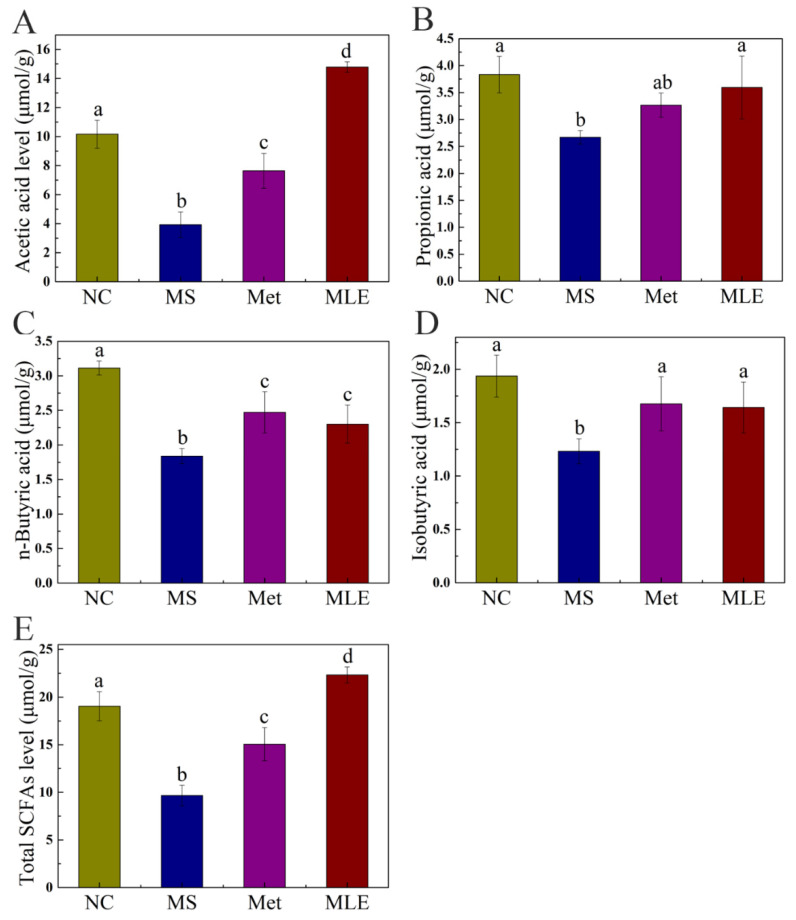
Effects of MLE on the types and contents of SCFAs in the gut microbiota of MS mice. After 15 weeks of intervention, GC was used to determine the types and contents of fecal SCFAs in the normal group (NC), model group (MS), metformin treatment group (Met), and MLE treatment group (MLE). (**A**) Acetic acid content in feces, (**B**) Propionic acid content in feces, (**C**) Butyric acid content in feces, (**D**) Isobutyric acid content in feces, (**E**) Total SCFAs content in feces. Results were analyzed by one-way ANOVA followed by Duncan test. Different letters (a–d) indicate values with statistically significant differences, “ab” indicates no significant difference from groups with letter “a” or “b”.

**Figure 11 molecules-27-05111-f011:**
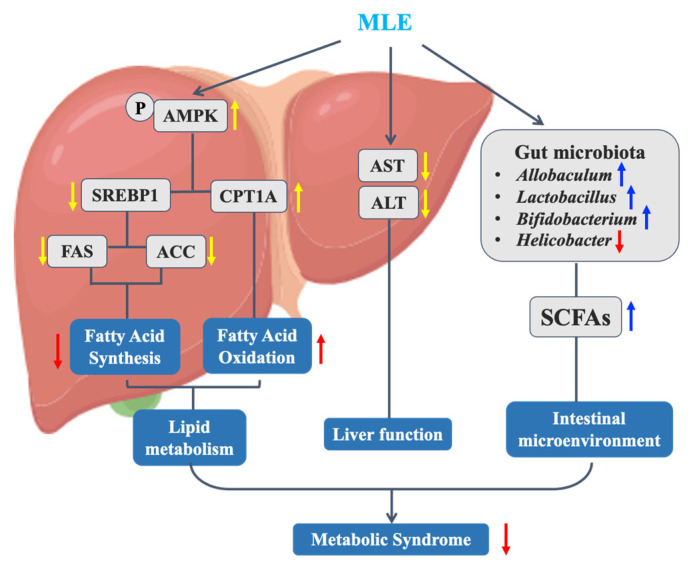
Mechanism of MLE improving metabolic syndrome in MS mice.

**Table 1 molecules-27-05111-t001:** The identified compounds in MLE by HPLC-MS/MS.

No.	RT (min)	Observed *m*/*z*	Calculated *m*/*z*	ΔMass (ppm)	Molecular Formula	Compound Name
1	3.74	339.0717	339.0711	1.77	C_15_H_16_O_9_	Cichoriin
2	4.17	353.088	353.0867	3.68	C_16_H_18_O_9_	Chlorogenic acid
3	5.53	609.1447	609.1450	−0.49	C_27_H_30_O_16_	Rutin
4	6.22	463.0862	463.0871	−1.94	C_21_H_20_O_12_	Hyperoside
5	7.01	447.0911	447.0922	−2.46	C_21_H_20_O_11_	Astragalin
6	7.35	505.0989	505.0977	2.38	C_21_H_20_O_11_	Quercetin

RT: Retention time in HPLC.

## Data Availability

The data presented in this study are available in Appendix A.

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
