# Peer review of "Mulberry Leaf Extract Improves Metabolic Syndrome by Alleviating Lipid Accumulation In Vitro and In Vivo"

_molecules, 2022, doi:10.3390/molecules27165111_

Round 1

Reviewer 1 Report

The manuscript describes the effects of Mulberry leaf extract (MLE) on lipid metabolism in vitro and in vivo and the additional effect of MLE on microbiota. I have some concerns regarding some data presentation.

1) Line 107, pag.3: “the viability of the cells administered with MLE with a concentration above 50 μM was significantly decreased compared to the control group” does not reflect data presented in figure 2A.

2) Line 209, pag.8: “MLE decreased the size and number of lipid vacuoles…” The authors must support this statement with the evaluation of cell diameter and cell number.

3) Leptin and Adiponectin results need to be discussed. Authors should explain how leptin levels increase after MLE treatment and Oil red values decreases.

Reviewer 2 Report

After reviewing the manuscript, I found critical concerns that must be clarified and addressed. Therefore, I recommend the authors consider the following points.

Major points:

Introduction section

  1. In the Introduction section, I recommend the authors mention additional pharmacological effects of flavonoids isolated from Morus alba L. with antimicrobial, antiviral, and anti-inflammatory properties. This information can be extracted from the recommended reference (DOI: 10.1016/j.jep.2019.112296). 

Materials and Methods section

  1. Please provide information about the amount of the used samples. 
  2. All experiments should be performed according to proper methodologies. Please add the appropriate citation for each experiment.  
  3. Most importantly, animal experiments with single doses of tested extracts or compounds are not acceptable. At least two doses have to be tested. The authors should clarify this point, where the in vivo results are based on single doses of the used extract. 

Minor points:

  1. I recommend the authors double-check the full text for grammatical and typing errors.

Author Response

Response to Comments from Reviewer 2

Dear Reviewer:

Thank you very much for your effort reviewing our manuscript entitled “Mulberry leaf extract improves metabolic syndrome by alleviating lipid accumulation in vitro and in vivo”. We have carefully addressed your comments and made revisions accordingly in our updated manuscript. The revisions have been marked as tracked changes in the manuscript, all the page and line numbers in this response reflect the revisions in the manuscript under “show revisions in balloons” mode. The point-to-point response to your comments are as follows:

After reviewing the manuscript, I found critical concerns that must be clarified and addressed. Therefore, I recommend the authors consider the following points.

Major points:

Introduction section

In the Introduction section, I recommend the authors mention additional pharmacological effects of flavonoids isolated from Morus alba L. with antimicrobial, antiviral, and anti-inflammatory properties. This information can be extracted from the recommended reference (DOI: 10.1016/j.jep.2019.112296).

Response: Thanks for your comment. According to your suggestion, we have provided additional information on the pharmacological effects of mulberry leaf flavonoids on Page 2, Lines 66-69. Especially, the literature you recommended was discussion on Page 2, Lines 68-69 and cited as Reference 9 in the updated manuscript. We also searched for some additional papers similar to the one you recommended for discussion and cited as References 7 and 10 in the updated manuscript.

Materials and Methods section

Please provide information about the amount of the used samples.

All experiments should be performed according to proper methodologies. Please add the appropriate citation for each experiment. 

Response: Thanks for your comment. According to your suggestion, we have also provided the specific number and the amount of samples used accordingly on Page 16, Line 606, Line 632 and Line 640.

We have also supplemented the citations of each experiment as follows: Reference 53: on Page 14, Lines 528-529 (for Section 4.2); Reference 54: Page 15, Lines 546-547 (for Section 4.3); Reference 55: Page 15, Lines 560-561 (for Section 4.4); Reference 56: on Page 15, Lines 568-569 (for Section 4.4); Reference 57: on Page 15, Lines 575-576 (for Section 4.5); Reference 58: on Page 15, Lines 584-585 (for Section 4.6); Reference 59: on Page 16, Lines 600-601 (for Section 4.7); Reference 60: on Page 16, Lines 614-615 (for Section 4.9) and on Page 16, Lines 640-641 (for Section 4.11); Reference 61: on Page 16, Lines 630-631 (for Section 4.10). We have also carefully checked the whole manuscript to correct similar issues in the text.

Most importantly, animal experiments with single doses of tested extracts or compounds are not acceptable. At least two doses have to be tested. The authors should clarify this point, where the in vivo results are based on single doses of the used extract.

Response: Thanks for your comment. We fully agree with the reviewer that animal studies for drug development require two or multiple doses, as one of the purposes of animal study for drug development is to provide potentially reasonable doses for future treatment. However, in the fields of functional food and food supplement studies, the requirements for multiple doses are less stringent. In vivo animal studies with single dose are acceptable, with examples including: Li et al. Polysaccharides from Callerya speciosa alleviate metabolic disorders and gut microbiota dysbiosis in diet-induced obese C57BL/6 mice. Food & Function 2022, DOI: 10.1039/D2FO00337F; Tian et al. Sulforaphane regulates glucose and lipid metabolisms in obese mice by restraining JNK and activating insulin and FGF21 signal pathways. J. Agric. Food Chem. 2021, 69, 13066−13079.; Luo et al. Antiobesity effect of flaxseed polysaccharide via inducing satiety due to leptin resistance removal and promoting lipid metabolism through the AMP-activated protein kinase (AMPK) signaling pathway. J. Agric. Food Chem. 2019, 67, 7040-7049. DOI: 10.1021/acs.jafc.9b02434. As far as we know, in the fields of functional food and food supplement research, it is more important to explore the active ingredients of edible plants and their mechanisms of action.

Actually, we previously applied for the multi-dose animal study to our Institutional Animal Care and Use Committee. However, the committee recommends that we conduct single-dose animal experiment based on our reasonable in vitro data available and the 3 ‘R’ principles (replacement, reduction, and refinement) in laboratory animal studies. We wish we had argued with the committee for allowing us to conduct multi-dose animal experiments, and we will argue with our committee as multi-dose animal experiments can provide more useful information in future studies. Fortunately, even with a single dose, we discovered the effects and mechanism of MLE. We attribute it to our solid investigation in in vitro studies and the attention paid in in vivo experiment design.

We apologize for not emphasizing that the purpose of this manuscript is to develop dietary supplements, we have provided this information in the Abstract (Page 1, Lines 21-22), Introduction (Page 2, Lines 94-95) and Discussion (Page 14, Lines 508-510) sections. We wish that the revised manuscript will meet the requirements of animal testing in fields of functional food and food supplement research.

Minor points:

I recommend the authors double-check the full text for grammatical and typing errors.

Response: Thanks for your comment. According to your suggestion, we double-checked the article for grammar, bibliography, and typing errors. The representative modifications were made on Page 2 Line 56, Line 72, Line 78, Page 3 Line 121, Page 4 Lines 147-148, Page 6 Line 200, Page 7 Lines 210-212, Line 215, Line 222, Page 9 Line 277, Lines 286, 293, 299; Page 10 Lines 320-321; Page 11 Line 349; Page 12 Lines 354-355, Lines 370, 384, 385; Page 13 Line 432, 456; and Page 14 Line 495, 497 and 501.

Round 2

Reviewer 1 Report

The authors replied to reviewer's comments and the manuscript was significantly improved

Reviewer 2 Report

The manuscript has been sufficiently improved.